# Public Space Behaviors and Intentions: The Role of Gender through the Window of Culture, Case of Kerman

**DOI:** 10.3390/bs12100388

**Published:** 2022-10-10

**Authors:** Aida Jalalkamali, Naciye Doratli

**Affiliations:** Department of Architecture, Eastern Mediterranean University, North Cyprus via Mersin 10, Famagusta 99628, Turkey

**Keywords:** gender, culture, public space behavior, Iran, traditional ideology

## Abstract

Urban public spaces are the spatial nexus of social interactions, activities, and expressions. Society manifests itself within these spaces through different lenses such as cultures and norms. The framework and restrictions related to gender-based behavior play a central role in the discourse about inclusivity and equity in urban public spaces when viewed through these lenses. There is, however, a gap in the literature that addresses how culture and gender influence public space behaviors and intentions within a traditional and modern cultural framework. The current study tests whether culture correlates with public space behavior on a neighborhood scale. A comparative study was conducted between two distinct neighborhoods in the city of Kerman, Iran. The two cases were selected due to their significant differences in how women use and interact with urban space. The effects of gender differences and perceived constraints on how residents manifest their behavior in public spaces were investigated. The study further investigated whether gender has a significant relationship with the level of appropriateness of certain public space behaviors and the intention to pursue them. The findings indicate significant cultural differences in both behavioral practice and behavioral intentions among the respondents. Furthermore, in traditional neighborhoods, the findings support significant gender-dependent differences in public space behavior, even though this gap is not apparent regarding intentions toward public space behaviors. The findings show a significantly greater disparity between traditional and modern districts in women’s perceived behavioral restrictions on personal expression.

## 1. Introduction

Public space behavior in relation to gender generally has been discussed in the fields of urban studies [1], gender studies [2], and feminist geography [3]. These fields focus on particular aspects of the relationship. First, the field of urban studies investigates differences between men’s and women’s spatial behavior in urban public spaces through different perspectives such as privacy [4], safety [5], characteristics of the built environment [6], and the right to the city [7], to name a few. Researchers in this field often explore the concept of space from a social point of view and explore how urban spaces are socially constructed and, consequently, gendered [8,9].

Urban studies literature often addresses gender under the umbrella of inclusivity of access, activities, and the right to the city [10,11]. Furthermore, the relation between women’s subordination and the levels of their presence and activeness in urban public spaces is central to the field [12]. These studies are often case-based and use mapping techniques to meet the goals of the study [13]. Moreover, urban studies can have alternate foci related to fear of crime and vulnerability which affect women and their presence and activity levels in urban public spaces [14,15,16,17,18,19,20,21,22,23,24,25,26,27]. Another aspect of urban studies focuses on the physical characteristics of urban public spaces, which can influence gender-based behavior [28,29,30]. Some culture-based studies evaluate how either a specific cultural set affects the physical arrangement of public and private spaces in a certain context, or how women modify their behaviors according to the normalized space definition [4,9,31,32,33,34,35,36,37].

Second, in gender studies, researchers negotiate gender rights and explore how women have had unequal access to urban public spaces throughout history [7]. The ongoing debates in these studies are based on gender-role theory. According to this theory, men’s and women’s behaviors are the result of the interactions between individuals and their environments. These appropriate gender roles are specified according to a society’s values and beliefs about differences between sexes [38]. Therefore, this process imposes domestic roles on women while allowing men to have more active and public roles. Gender research connects these concepts to urban studies by introducing two realms: public and private. These realms have physical representations in cities; all of the urban public spaces, industrial areas, and productive parts of the city represent the public realm, whereas the domestic realm of the city, rural areas, and houses represent the private realm. These distinctions suggest that because women are more associated with domestic duties, they will be held back from being present and active in urban public spaces [12,39,40].

The third field of study is conducted by feminist geographers, who focus on the socio-economic aspects of gender relations, negotiating gender as a political feature, dealing with gender inequality in the use of urban public spaces, and establishing equal physical access to these spaces. Some feminist research is more involved in the urban planning process. These perspectives’ planning practices emphasize the critical role of women’s participation, involvement, and inclusivity through the lens of everyday life, which is frequently constrained by social norms [41]. The male-oriented nature of planning and design, in many cases, has the tendency to exclude women from these spaces [42,43]. Gender segregation and internalized patriarchy are as old as human history [44]—the detailed explanation of which goes beyond the scope of the current paper. In general, gender segregation in the labor force has caused men to become more associated with outside, public, moneymaking jobs, and women to more often perform domestic, private, and unpaid housework [44]. This inequality was also intensified by the process of urban planning. In the United States, for instance, cities grew into two main zones where the gender balance was tilted: the city center, where the most productive activities were typically performed by men, and suburban zones, where more domestic and private activities were undertaken by women [45]. Furthermore, how urban structures limit women’s opportunities, reinforce unequal access to resources, and direct urban space usage has been studied [39,46,47,48,49,50,51,52,53,54,55,56].

Furthermore, two main theories investigate behavioral patterns based on cultural differences: Hofstede’s cultural dimensions theory [57,58,59] and Schneider’s ASA (attraction–selection–attrition) model [60,61]. Each theory predicts how people from different regions and cultures will behave differently, but neither predicts cultural influence over gender-based behavioral differences. These theories, however, are broad and do not propose ideas regarding public space behaviors.

Reviewing the existing body of literature shows a myriad of studies targeting some of the aforementioned topics, such as the importance of mixed gender presence in urban public spaces [13], gender-based violence [62], sense of safety [5,63], challenging social norms [64], gender equity [65], and gender nonconformity [66], to name a few. Ideas of Hofstede and Schneider have been also interpreted with regard to urban public space behavior [67,68,69]—albeit very few and limited in scope. However, studies exploring empirical and theoretical dimensions regarding the influence of gender and culture on urban public space behavior seem to require more attention. Therefore, addressing how culture and gender are related to public space behavior, the current study tries to fill the following gaps:Urban studies do not sufficiently address how exactly culture is related to behavioral studies and more specifically the way men and women behave differently in urban public spaces.Gender studies and feminist geography—those negotiating gender roles, equal rights, and access to urban public spaces—are not adequately concerned with cultural influence on the relationships that men and women establish with urban public spaces.Cross-cultural models, which study behavioral differences, lack attention to gender differences in behavioral patterns.

This paper examines how culture and gender shape individuals’ public space behaviors and the importance they give to different divisions of behaviors in urban public spaces within the framework of social construction theory and social cognition theory. This study tests, in two sets of culturally different neighborhoods, how men and women differ in their comfort level when pursuing certain behaviors in urban public spaces. Personal expression, social interaction, and physical activities are identified as the three main divisions of public space behavior in this study. The study also tests the extent to which practicing these behaviors is appropriate for men and women in each neighborhood. By doing so, this study addresses a gap in current knowledge regarding how cultural differences and socio-cultural factors shape gender stereotypes and apply them to public space behavior.

The sample consists of 400 residents divided between a modern and a traditional district in Kerman, Iran. These districts differ in terms of their level of traditionalism. In the context of this city, traditionalism means upholding more conservative and Islamic norms. While in Islamic education and tradition, public space behavior needs to be modest, it is far more restricted for women than for men, especially when perceived from the perspective of more modern and secular schools of thought. In addition, this study aims to gain a better understanding of Iranian Muslim women’s presence and socio-spatial behaviors in urban public spaces. Although the same process of Islamic education and advertising has affected all neighborhoods during the previous 40 years, the residents’ public space behaviors appear to vary from one neighborhood to the other. This study demonstrates how public space behavior can become a gendered act and how culture can affect this act through processes of social construction and social cognition. Ultimately, this research discerns whether public space behavior differs according to culture or gender at the neighborhood scale.

## 2. Theories and Hypotheses

### 2.1. The Social Construction of Gender and Public Space Behavior

Public space behavior needs to be understood as a gendered act in order to explore how the process of social construction affects any particular gendered behavior. Differential socialization is a result of androcentric and patriarchal culture as well as the general normalization of stereotypes and roles in communities. According to this theory, gender is a socially constructed phenomenon [70,71,72,73,74]. The process of socialization guides boys and girls to acquire their appropriate gender-typed behaviors [75,76]. In this process, first, the family plays the main role for boys and girls, teaching them which particular behaviors are appropriate. Families, by developing strong preferences for gender-specific activities, accelerate the process of applying gender stereotypes. In many cases, these norms are often reinforced during formal education that indoctrinates boys toward being heroes and girls to be more passive and compliant [76,77]; although these differences might be as simple as color coding or media, they have significant long-term effects [76]. These intrinsic inequalities that are often embedded in the structure of educational systems [78,79] amplify gender-based socialization processes [80]. The alienation of women through these socialization processes has the tendency to extend into society as a whole, impacting their rights and even how they imagine themselves as part of society [81].

Best and William [82] presented stereotypes and gender roles as part of the main factors that contribute to the development of cultural differences that influence the behaviors of males and females. Once stereotypes are established, they serve as socialization models, encouraging boys to become adventurous and independent and girls to become cooperative and nurturing. These common stereotypes for boys and girls can be extended to stereotypical behavior in public spaces as well [82]. Franck and Paxson [12] tried to explain how some of these stereotypical behaviors affect the spatial behavior of boys and girls in urban public spaces. They indicated that girls are expected from an early age to be less physically active, more fearful, and less exploratory than boys are. Franck and Paxson continued that the spatial range of girls’ activities is smaller than that of boys’, and they are less likely to be involved in activities that manipulate the environment. This is mostly because of the socialization process, where girls learn to use less space and be gentler. Learning appropriate public behavior according to gender is not a different process than learning other behaviors. Adult women apply the same rules to their behavior when they use public spaces [12].

Observation and role models are two other instruments in the process of social learning. The way people observe and imitate various same-sex role models provided by society also influences public space behavior [83]. According to Perry [83], boys observe which types of public activities and behaviors are for boys and girls learn which are for girls by watching others. Initially, girls look up to their mothers and boys to their fathers. The process continues with other same-sex role models, such as peers, teachers, and media personalities. Stereotypical gender-based public behavior is seen in the behavior and activities of a society’s men and women.

Leaper and Friedman [84] asserted that one of the main sources for acquiring cultural information about appropriate gender behavior is the mass media. Thompson and Zerbinos [85] and Emons, et al. [86] talked about pervasive gender stereotypes in many American television series. One stereotypical example that can be observed on television is that boys play football and girls are cheerleaders. Although this simple example is not pervasive or unique, it shows how the appropriateness of behavior can be influenced by the media. If boys see their father going out in public spaces together with other men to exercise, run, or do other physical activities, it is more probable that they will acquire similar habits. Meanwhile, when a girl sees that her mother, sisters, or girlfriends would not choose to exercise in a public space, she is also less likely to do so.

Context and culture are coordinated in the process of socialization to achieve the appropriate gender-based behavior, including public behavior, through social norms [87]. These norms are the unwritten rules for how men and women should act and behave in public. As Marcus et al. [88] found, “social norm[s] emphasize the importance of shared expectations or informal rules among a set of people… as to how people should behave”. Bicchieri [89] and Heise [90] defined social norms another way: when certain patterns of behavior repeat, they become the unwritten rules of social norms or standards of behavior for people to reference. Marcus, Harper, Brodbeck, and Page [88] indicated how social norms affect the practice of public behavior and vice versa, meaning any behavioral practice in public space creates certain behavioral norms for public spaces.

These regular patterns of public space behavior, which can be seen as the manifestation of norms, are also applicable to gender-based public behavior. Gender norms are related specifically to gender differences [88]; Marcus, Harper, Brodbeck, and Page [88] defined them as the “informal rules and shared social expectations that distinguish expected behavior based on gender”. Throughout the learning process, if these gender-based public norms are ignored, then individuals may face sanctions from society (Figure 1).

### 2.2. Social Cognitive Theory and Public Space Behavior

Apart from the social construction theory, which shows the relation between culture and gender-appropriate public behavior, the social-cognitive process of gender construction completes the process of cultural influence over gender-based public behavior [92].

In addition to the norms that boys and girls acquire from their social environments, they also have self-realization that gives them individuality. As Banerjee [93] stated, “once children have begun to internalize the standards of behavior appropriate for males and females, based on the social experiences described for them, their behavior is no longer just dependent on external rewards or punishments. Rather, they become capable of directing their behavior in such a way as to satisfy their internalized standards”.

These internalized standards are a source of information that helps to define gender identity. That which people determine to be appropriate for their gender will be accepted as part of their identity. The process of individuals becoming gendered in society is defined by schema theory. “Gender schema theory is a process where the schemas are active, facilitating a relationship between the child’s thoughts, behavior, and in turn shaping the development of their gender and attitudes towards the self” [94]. The behaviors with which children become gender-typed—choosing specific characteristics and beliefs that are considered appropriate for one gender but not the other—often have to pass through gender-based schematic processing. Through socialization, this process sorts gender-related information relevant to the self—like activities, attitudes, or behaviors—according to what people should and should not present given their sex (see Bussey and Bandura [95], Martin et al. [96], and Turner [97]). This process is complete when they have ‘self-evaluation’, as Bem [98] reasoned: children have a category of ‘own-sex schema’. They will start to compare themselves to their gender schema, evaluating their preferences, attitudes, behaviors, and personal attributes. At this stage, children form their values, create certain beliefs, and judge their own and others’ behaviors and attitudes as right or wrong based on the gender identity (own-sex schema) that they have gained from their society [84]. This process includes their self-expression and behavior in public spaces. Figure 2 presents an example of this process, which can be adjusted to the context of a culture.

Regarding social cognitive practices, Bussey and Bandura [95] argue that development and differentiation of gender emerge through bidirectional and mutual influences of three factors: personal, behavioral, and environment. The personal factor relates to self-realization, self-regulation based on certain beliefs, cultural norms, evaluation, judgment, and decision-making (e.g., if I am a good Muslim woman, I may not chat with a strange man when in a public space). Behavioral factors include behaviors or actions linked to gender (e.g., men presenting more aggressive behaviors in public spaces). Environmental factors relate to social influences that are experienced in everyday interactions. The approach emphasizes the importance of learning gendered information, or in this case, gender-appropriate public behavior, through context (Figure 3).

### 2.3. Culture, Gender, and Public Space Behavior in Traditional-Islamic Context

While Cronk [99] proposed that behavioral patterns change from one cultural cluster to another, in Iran, public space behavior is expected to differ from one neighborhood to another due to differing degrees of exposure to Islamic culture. Current Islamic culture in Iran needs to be reviewed to determine if its effect on public space behavior is a gendered act or not.

In Islamic thought, activities that negatively affect the good ethics of society should be prohibited, and activities that make it stronger should be supported. Guarding the ethical aspects of society is considered a duty of all. The Surah Nor (Quran 24: 1–64, Oxford World’s Classics edition [100]) explains the duty of men and women when interacting:Men and women should not look at each other directly.Men and women should be clean and honest.Women should cover themselves; they should conceal their beauty and makeup in the presence of strangers, lest they attract and seduce them.

In Islam, both men and women are asked to be modest and behave cautiously; however, women are the main target in this text. Through their actions, women carry more of the responsibility of protecting the morality of Islamic society. For example, the Surah Ahzab, Sentence 34, (Quran 33: 34–35, Oxford World’s Classics edition [100]) provides guidance regarding the presence of women in public spaces and reinforces the appropriate behavior: “Women, stay in your houses, and don’t go out like the times before you became Muslim, to show yourself with your beauty.” As Azani and Zal [101] stated, it is not appropriate for Muslim women to go out and have fun because it is counter to their main duties. They claim that the woman’s duty is to stay at home, take care of the children, and remain away from public eyes as much as possible; therefore, her presence outside the home is not advised [101].

Meanwhile, Islam suggests frameworks for appropriate behavior and speaking manners for men and women. As stated in Islamic texts, men and women should avoid any kind of activity that causes temptation. They should have a modest appearance and act with dignity when they want to be present in public. Women should be more careful with their actions and behaviors in public because it is easier for them to jeopardize the dignity of society. For instance, Motahari [102], one of the most famous pioneers of modernizing Islamic thought after the 1979 Iranian Revolution, asserted that women may not walk in a way to show off and draw people’s attention. Through these perspectives, the mere presence of a woman in public is considered provocative and requires legislative intervention [102].

Some other narratives in Islam claim that the voice of a woman is sexually provocative and forbidden to be heard by men. Although the cultural progressiveness of Iranian women has eradicated these extremist approaches in most parts, still, women are expected to speak very modestly and in monotone—without any expression of joy or happiness [101]. Through the window of Islamic education, intense physical activities, like sports, dancing, or any kind of exciting behavior, are not recommended for women [103]. These types of activities are condemned for women, and those women who perform them are not considered to be well-reputed Muslims [103].

Accordingly, public space behavior in Islamic ideology is a gendered act in all three divisions. First, for personal expression, women should cover themselves in public, avoid applying makeup, and not raise their voices (laughing or talking loudly). Second, in social interaction, women are asked not to have any interactions with the opposite sex other than their family members. Third, in physical activities, women are asked to be less physically active in public spaces.

Bagheri [9] argued that, in Islamic culture, gender inequality not only exists in public space behavior but is also reinforced and legitimized by social norms in more traditional contexts in Iran. Furthermore, Zakaiee [104] exposed a positive relationship between women’s behavior in public space and the culture of a neighborhood. The cultural orientations of each district support a certain process of socialization and social cognition; the perception of these gender inequalities becomes internalized differently by the men than by the women living in each district. Therefore, it may be expected that there would be a higher level of gender inequality in public space behavior in traditional neighborhoods than in neighborhoods with a strong modern influence.

Due to the traditional Islamic culture that emphasizes gender-role differences [105], it is expected that men in these cultural sets are freer in their range of public space behavior. Men can be more assertive, competitive, and tough-acting in public, while women are supposed to be less apparent in public, be gentle in their behaviors, and avoid conflict and risk-taking behaviors. Witt and Wood [106] elaborated that when a behavior is considered to be gendered, individuals will adapt their abilities to pursue that behavior due to self-concept and gender stereotypes. Therefore, it may be expected that women in traditional neighborhoods perceive more restrictions on public behavior compared to their male counterparts. Given this information, the study proposes the first three of six hypotheses (Figure 4).

**Hypothesis** **1a**(**H1a**)**:** *The likelihood that women perceive more behavioral restrictions in personal expression than men will be greater in traditional neighborhoods than in modern neighborhoods.*

**Hypothesis** **1b**(**H1b**)**:** *The likelihood that women perceive more behavioral restrictions in social interactions than men will be greater in traditional neighborhoods than in modern neighborhoods.*

**Hypothesis** **1c**(**H1c**)**:** *The likelihood that women perceive more behavioral restrictions in physical activities than men will be greater in traditional neighborhoods than in modern neighborhoods.*

### 2.4. Behavioral Intentions in Urban Public Spaces

The theory of planned behavior proposed by Ajzen [107] describes three behavioral intentions—attitude, subjective norm, and perceived behavioral control—that shape an individual’s behavior. This theory has been applied worldwide in various fields of study, such as health science, business, management, anthropology, and sociology. According to the theory, if people positively evaluate a named behavior (e.g., it aligns with their self-concept), and they see that current socio-cultural norms—along with their significant others (family, friends, and peers)—are not opposed to the behavior, then their intention to perform that particular behavior may increase.

In many studies, it has been confirmed that there is a high correlation between attitudes, subjective norms, and behavioral intentions, and, consequently, with behavioral performance as well [108,109]. The first two factors of planned behavior are the basis for the intention to perform the actual behavior. However, in this study, the examination is based on behavioral control, which refers to the perceived appropriateness or inappropriateness of performing the behavior (in this case, public space behavior).

The desire to pursue appropriate public space behavior does not seem to be consistent across genders or cultures. In their study on entrepreneurial intentions, Drnovsek and Erikson [110] proposed that the perceived appropriateness of the action, along with the perceived ability to implement the action, shape behavioral intentions. In another study, Hassan and Shiu [111] discussed how behavioral intentions are affected by perceived abilities as personal factors and culture as a social factor. Based on these concepts, this study anticipates that gender itself and culturally shaped gendered behaviors in public space can have a significant relationship with the level of appropriateness and the intention to perform a behavior in urban public spaces. If a public space behavior of personal expression, social interaction, or physical activity is not considered to be desirable or appropriate for a woman in traditional Islamic culture, even a woman who feels those restrictions are not relevant to her may choose not to perform the behavior. In order to examine the relationship between the perceived appropriateness of culture, gender, and public behaviors, the study proposes (Figure 5):

**Hypothesis** **2a**(**H2a**)**:** *The significance of the relationship between gender and the level of appropriateness in personal expressions will be higher in traditional neighborhoods than in modern neighborhoods.*

**Hypothesis** **2b**(**H2b**)**:** *The significance of the relationship between gender and level of appropriateness in social interaction will be higher in traditional neighborhoods than in modern neighborhoods.*

**Hypothesis** **2c**(**H2c**)**:** *The significance of the relationship between gender and the level of appropriateness in physical activities will be higher in traditional neighborhoods than in modern neighborhoods.*

## 3. Materials and Methods

### 3.1. Case Study

Kerman, Iran, is located near the Lut Desert. The city, with its unique history and special socio-cultural geography, hosts areas that remain traditional and conservative. In these parts of the city, Islamic culture has been rooted since before the Islamic Revolution in Iran. Nevertheless, Islamic influence became stronger following the revolution. More recently developed parts of the city were shaped and influenced during the Pahlavi dynasty (1925–79), making them more modern with more secular and open-minded residents. The modern areas have continued to extend into peripheral neighborhoods, yet these two socio-culturally contrasting parts of the city have stayed geographically distinct. The historic and traditional parts of the city are located in the north, and the more newly developed and culturally modern areas are located in the south of the city.

Two different socio-cultural districts of Kerman were selected for the sample. One northern district is composed of five traditional neighborhoods, and one southern district comprises five modern neighborhoods (Figure 6). The districts were selected according to the degree to which they are exposed to Islamic culture—for example, the number of mosques they have and the number of cultural centers where Islamic ideologies are promoted. In addition, these two districts reflect distinct social norms, which are reflected in people’s clothing styles and behavior in public. The sample for the study includes 400 residents of Kerman (215 from traditional neighborhoods and 185 from modern neighborhoods).

Altogether, the traditional neighborhoods have 35 mosques and host the two main Islamic cultural centers. One of these neighborhoods contains the Islamic soldiers’ graveyard, which also houses the Islamic war museum. Another neighborhood of the traditional district hosts the most dominant government-run Islamic cultural center (مصلی) in the city. These neighborhoods emanate an Islamic atmosphere where all behaviors and movements are watched and any act out of the social norm garners attention. Women in these traditional neighborhoods behave and dress more conservatively and modestly (Figure 7).

The modern neighborhoods, altogether, host only four mosques, and, at the same time, most of the entertainment venues and shopping malls are located in these neighborhoods. Even though the official Islamic regulations are in force throughout the city, the social norms in the modern district overrule the Islamic cultural expectations. Residents display their preferences in the way they dress, socialize, and are physically active in public (Figure 8). It is evident that people in these neighborhoods act more liberally, and their behaviors are not being supervised by others.

### 3.2. Sample and Data Collection

The target population of this study was adult men and women residing in two districts of Kerman, Iran (one traditional and one modern). The targeted populations consist of people who permanently live in these neighborhoods. According to the Statistical Center of Iran [112], the traditional district has a population of 3986, of whom 2150 are over 18 years of age. The modern district has a population of 2654, of whom 1846 are over 18 years of age. The minimum target sample size was determined following the guideline table recommended by Bartlett et al. [113]. Accordingly, to reach a 95% confidence interval, the minimum was set at 198 for the traditional neighborhood and 183 for the modern neighborhood [113]. Finally, 215 people from the traditional district and 185 people from the modern district voluntarily participated in the survey.

It must be noted that the very definition of what constitutes public space is subject to debate, uncertainty, and contextual preferences [114]. In this study, the data were collected in two urban parks in the evenings. The data collection was repeated until the aforementioned desirable sample size was reached. It must be noted that although the survey was approved by the ethics committee board of the university, conducting the survey on the actual ground of the city—particularly when it targets gender and culture-related issues—was difficult due to the sensitivity of the government on such issues. Accordingly, parks provided a more suitable environment for the data collection because the nature of their public life is more leisurely, stationary, and relaxed. The study tried to make the sample a reasonable representation of the target population, although generalization was not the primary goal and would be best avoided. The main purpose of this study was to determine whether public space behavior has a significant relationship with culture and gender. Any correlation that culture and gender have with public space behavior can be generalized to other contexts as long as the level of cultural difference is significant between at least two groups of the study.

### 3.3. Variables

The study assessed the perceptions of three divisions of public space behavior on a five-point Likert scale: (1) personal expression, (2) social interaction, and (3) physical activity. In the questionnaire, respondents were asked to rate their comfort in performing each behavior in public space, with “1” being “always” and “5” being “never.” Therefore, respondents scored highly if they did not feel comfortable performing the behavior publicly.

Personal expression was assessed via four behaviors: (1) going to urban public spaces wearing clothing of personal choice; (2) laughing or speaking aloud with friends in urban public spaces; (3) expressing personal feelings of pleasure or excitement in urban public spaces by expressive physical actions; and (4) going to an urban public space with hair color, hairstyle, or makeup of personal choice. It is critical to note that many of these entries are designed to highlight different degrees of intrinsic gender differences in the context. These items might be considered out of the ordinary—or not—in many public contexts and should not be overgeneralized (see [115]). For instance, items 1 and 4 represent the differences in the most basic forms of passive expression, that is, clothing choice. This is forced by the governing body via means such as the morality police and is also reinforced by religious norms (see [116]). Just as the norms might limit clothing and style, many use these mediums to fight back and challenge those very norms. Item 3 aims to address a more radical expression that might not be identifiable through other items. Nevertheless, the limitations of these items in addressing public space behavior are outlined at the end of this paper.

Social interaction was assessed via four behaviors: (1) joining social groups in an urban public space; (2) spending time with a mixed-gender group of friends in an urban public space; (3) walking in an urban public space with a person of the opposite sex; and (4) making friends and getting to know new people in an urban public space. Physical activity was assessed by four behaviors as well: (1) jogging in an urban public space; (2) running in an urban public space; (3) joining and playing games (e.g., football, ping-pong, or basketball) in an urban public space; and (4) using sports facilities in public parks.

The survey also examined whether differences exist between the residents of the modern and traditional districts concerning gender and the perceived appropriateness of named behaviors in public space. The study measured the perceived appropriateness of each public space behavior in the same three categories on a five-point Likert scale: (1) personal expression, (2) social interaction, and (3) physical activity. Respondents were asked to rate the appropriateness of named behaviors in urban public space, with “1” being “strongly disagree” and “5” being “strongly agree”. Therefore, respondents scored lowly if they did not perceive the behavior to be appropriate public space behavior.

### 3.4. Data Validation

The data were verified in two steps. First, the construct validation was controlled via principal component analysis (PCA) [117]. This step explored whether the composed latent variables corresponded to their intended targets [118]. The factor loadings confirmed the isolation of three main factors in two divisions with an eigenvalue greater than one (Figure 9). Overall, 69.3% of the variance in the traditional neighborhood and 68.5% in the modern neighborhood were explained by the six isolated factors in two sets of questionnaires.

The study originally assumed that the participants from the two districts had different cultural contexts; therefore, an analysis of variance (ANOVA) was conducted to explore whether the assumption was valid [119]. ANOVA helps explore central significant tendencies among different target groups within a dataset [120]. ANOVA is often used in studies where gender is among the independent variables [121,122,123].

## 4. Results

### 4.1. Evaluating the Relationship between Culture and Behavior in Public Space

The ANOVA step did not directly explore the validity of the two sets of hypotheses but established a basis for them by showing the significance of differences among the participants from the two districts.

Table 1 displays the structural model for the study, though it considers only cultural differences between the two districts. Cultural differences are significant between these districts. The model shows culture correlating with public space behaviors at their performance levels. With an F value of 6.668, perceived behavioral restrictions in personal expression are significant for men in both modern and traditional neighborhoods; they are also significant for women in both modern and traditional neighborhoods, with an F value of 31.940. These results demonstrate that the disparity between women in modern and traditional neighborhoods is much greater than that between men in these two cultural sets and show that women perceive more behavioral restrictions than men do. In the second division of behavioral performance, perceived behavioral restrictions for social interaction are again significantly different between the men (F value = 18.507) and women (F value = 26.810) of these two cultural sets. In the third category of perceived behavioral restrictions for physical activities, cultural differences are not significant for men (F value = 1.198) between these two cultural sets, though they are still significant for women (F value = 8.879) when comparing the traditional and modern districts. This reveals that men in both modern and traditional districts do not experience restrictions regarding their performance of these behaviors in urban public spaces.

In the second stage, the study checked whether culture also has a significant correlation with behavioral intentions in urban public spaces. The results demonstrate that significant differences exist in the perceived level of behavioral appropriateness between the modern and traditional cultural sets for the three divisions of personal, social, and physical. The level of behavioral appropriateness for personal expression among the men in the modern and traditional neighborhoods scored an F value of 17.100, and for women, the F value was 26.810. Between the two districts, the level of behavioral appropriateness for social interactions attained an F value of 24.342 among men and an F value of 30.088 among women. Regarding physical activities, the level of behavioral appropriateness also showed significant differences for both men (F value = 5.699) and women (F value = 8.976) between the modern and traditional cultural sets.

### 4.2. Gender’s Effect on Practicing Public Space Behaviors (Testing First Three Hypotheses)

Following Figure 4 (Model 1), the study tested hypotheses H1a through H1c, examining the mediating effect of gender on the perceived level of behavioral restriction for the three divisions of personal, social, and physical public space behavior. Table 2 displays the structural model coefficient and the F values for Model 1, providing comparisons of gender-based differences across the two districts.

First, as the study hypothesized, the perceived behavioral restriction on personal expression is recognized as significantly more important for women than for men in traditional districts (F value = 10.058). A similar relationship exists between the perceived behavioral restrictions on social interaction with an F value of 11.089 and the perceived behavioral restrictions on physical activities with an F value of 8.054. In the traditional district, there is a significant gender disparity in all three divisions of practicing urban public space behaviors. On the other hand, the results showed that in the modern district there are no significant gender differences in the way men and women perceive behavioral restrictions in personal expression, social interaction, and physical activities. The results in each modern and traditional district confirmed that gender affects public space behavior in this traditional district of Kerman, while this pattern was not significant in the modern district.

### 4.3. Gender’s Relations with Behavioral Intentions in Urban Public Spaces (Testing Second Three Hypotheses)

Following Figure 5 (Model 2), the study tested hypotheses H2a through H2c, examining the relationship between the perceived appropriateness of public space behavior and the intentions to pursue a public space behavior by considering the role of gender. Table 3 presents the structural model coefficient and the F values for Model 2, comparing the modern and traditional neighborhoods.

Concerning the direct effects of gender and the perceived appropriateness of intentions for public space behavior, the results show small, negative effects of gender in modern neighborhoods. However, gender has nearly no moderating effect (F value = 0.473) on the relationship between perception of personal expression and level of behavioral appropriateness and intentions toward public space behavior, which rejects H2a for the traditional neighborhood. Gender has a negative moderating effect on the relationship between perceived social interactions and the level of behavioral appropriateness and intentions toward public space behavior, rejecting H2b with an F value of 0.824. As the comparisons show, no significant differences (F value = 0.544) appeared between men and women in traditional neighborhoods regarding their perception of physical activities and the level of behavioral appropriateness, which fails to support H2c.

## 5. Discussion

In examining how culture and gender shape life in the urban public spaces of Kerman, Iran, questionnaire findings indicate that culture has a significant and positive correlation with public space behavior, while gender’s influence has mixed results.

Both men and women in traditional neighborhoods perceived significantly more restrictions regarding their public behaviors in all three areas of personal expression, social interaction, and physical activity compared with their counterparts in modern neighborhoods. This pattern was repeated when it came to the perceived appropriateness of public space behavior; both men and women in traditional neighborhoods rated public space behaviors as significantly less appropriate for them to pursue than residents in modern neighborhoods indicated. The Islamic culture of traditional neighborhoods restricts men’s and women’s public space behavior and their intentions toward it when compared to modern neighborhoods. However, the results present a significantly greater disparity in women’s perceived behavioral restrictions to personal expression between traditional and modern districts; women in traditional neighborhoods restrict themselves more in terms of personal expression than women in modern neighborhoods do. Similarly, yet less pronounced, men in traditional neighborhoods perceive more restrictions on personal expression than their modern counterparts do. The fact that this disparity still exists for women in both districts shows that social interaction is a less sensitive issue than personal expression. The disparity in the perceived behavioral restriction to physical activity is significant between women in the two cultural sets (F value = 8.879), but it is minimal between men in the two districts (F value = 1.198). This shows that men in traditional neighborhoods are not restricted from performing physical activities in public.

Evaluation of gender’s effectiveness on public space behavior was performed on two levels: performing public space behavior and the intentions toward public space behavior. The results were interesting and contradictory. In the first part of the analysis, where H1a, H1b, and H1c were examined, the study confirmed that public space behavior—in the three divisions of personal expression, social interaction, and physical activity—is significantly a gendered act in traditional neighborhoods; the results support all three first hypotheses. However, when the research shifted to the moderating effect of gender on the relationship between the perceived appropriateness of public space behavior and the intentions to pursue public space behavior, the results revealed the very weak effect of gender on this relationship for all three domains in traditional neighborhoods. The contradiction is that public space performances are being gendered while the intentions toward their performance are not gendered in traditional neighborhoods. Because behavioral patterns are generally aligned with the intentions behind them [124], this contradictory finding is the most substantial result of this research. In most cases, it is the behavioral intention that forms the behavioral performance.

This mismatch reveals information about socio-cultural factors in the Islamic neighborhoods of Kerman. Certain behaviors are the result of internal factors (self-realization and identity) depending on what the individual perceives to be appropriate. Concurrently, external factors (cultural and social norms) reward appropriate behaviors and sanction unacceptable behaviors. Therefore, a possible explanation for this contradiction in traditional neighborhoods could be that either internal or external factors influence public space behavior, while the latter factors influence the intentions toward performing behaviors.

According to the findings, women’s public space behaviors in traditional neighborhoods are more restricted than men’s in all three divisions of personal expression, social interaction, and physical activity. This confirms that Islamic culture and local social norms limit women’s public space behaviors. However, when women in these neighborhoods were asked about their perception of the appropriateness of named public space behaviors, they saw themselves as approximately equal to men in their own socio-cultural context. Therefore, they are not totally under the influence of contextual culture; rather, they attribute individual value to certain behaviors.

It seems some women may have behaved differently if their contexts were different. Referring to one study wherein Bagheri [9] interviewed women in more traditional neighborhoods of Tehran, Iran, the author realized that whenever women wanted to have more social freedom—such as going out with friends of the opposite sex or wearing clothes and makeup of their personal choice in urban public spaces—they went to modern neighborhoods. In another study of various types of Tehran’s public spaces and the frequency of women being present in them, Zakaiee [104] realized that to feel less restricted in terms of their behavior, women chose to go far outside of urban public space by taking part in recreational activities such as hiking and mountaineering. In this regard, the restrictions that are imposed on women repel them from taking part in urban public space that is not only policed by governmental regulations, but also by the intrinsic norms of some sections of society.

These studies suggest that contextual culture is no longer the only definer of gendered values and behaviors. Nowadays, most people have access to the internet; therefore, diverse media is accessible and influential. The flow of information can reach beyond geographic borders, and Islamic culture cannot remain dominant. Women in these regions are gradually gaining equal status in terms of their presence in and use of urban public spaces; however, it may take future generations to overcome the contextual constraints that are currently in place.

## 6. Conclusions

This study endeavored to examine how culture and gender shape two aspects of life in urban public spaces: individual perceptions of restrictions on behavioral practice and intentions toward the perceived level of behavioral appropriateness. Urban public spaces of a city function as the main containers of production and consumption of gendered norms and identities.

However, in the process of gendering, culture constantly plays the main role. Concurrently, gendered embodiment and the socio-cultural reality of daily activity shape urban public spaces. However, the interdependence of urban public space, gender, and culture is not fully recognized in the numerous related efforts from urban social policy, urban studies, gender studies, and cultural studies; the concepts of urban, gender, culture, and behavior are primarily studied in separate fields of study.

What we see in Iran today—and has been demonstrated here—is a conflict between the social construction of gender and social cognition. Although gender norms are pushed through different media from an early age, women, in most parts, have not internalized these differences in their gender identity. What is perceived as appropriate public behavior is going through a paradigm shift. The rejection of Hypothesis 2 (a, b, and c) illustrates this change: where the perceived gender-based differences still partially exist (in the traditional neighborhood shown in Hypothesis 1), these differences are disappearing in regard to personal expressions, social interactions, and physical activities. The recent movement of “woman, life, freedom” and the fight against forced dress codes have shown the tendency for reform in social cognition of gender-based behavior.

This study tried to establish more coalesced research by bringing three rarely integrated disciplines into a dialog. The proposed model systematically intertwines urban and gender studies with cross-cultural understanding to explore behavioral differences between men and women in public space. The model was verified via a case study and survey that allowed a close examination of culture and gender’s influence on public behavior.

The results indicate that both culture and gender have a significant relationship with public space behavior, triggering different behaviors in the men and women of the two cultural sets. Moreover, this application on the specific cultural sets revealed a discrepancy despite the pair of districts being part of the same city with the same rules and regulations, even if different interpretations were made. This emphasizes the importance of evaluating public space behaviors within their context and culture, and it raises awareness of the need to consider these issues more often in urban studies. Proposing a policy to equalize gender rights—even in a single neighborhood of a city—requires an intricate approach.

In the end, it must be noted that the current study has certain limitations regarding scope, sample size, and area of analysis. Moreover, the contextual preferences and limitations of the country need to be considered in the interpretation of the findings.

## Figures and Tables

**Figure 1 behavsci-12-00388-f001:**
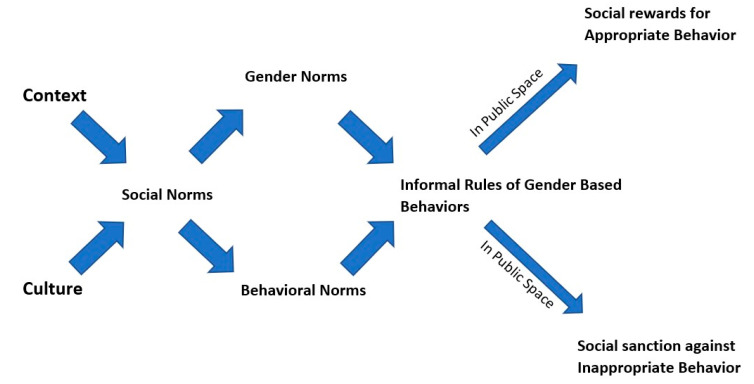
A diagram of how appropriate behaviors are reinforced for men and women according to current social norms. Adapted from Shaffer and Kipp [91].

**Figure 2 behavsci-12-00388-f002:**
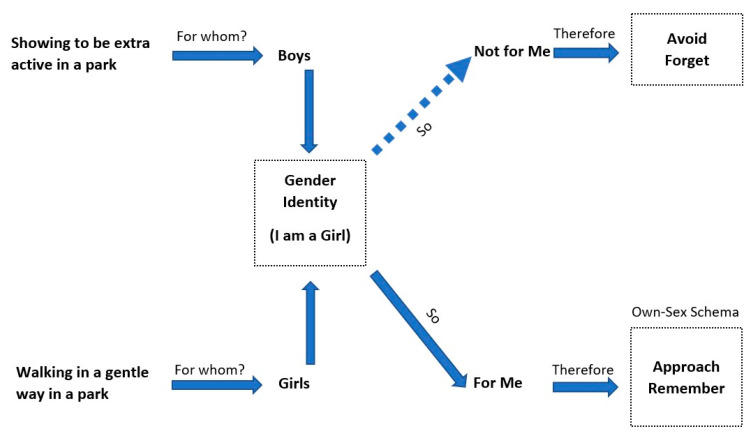
Gender schema in action: a diagram of categorizing appropriate behaviors in public space. Adapted from Shaffer and Kipp [91].

**Figure 3 behavsci-12-00388-f003:**
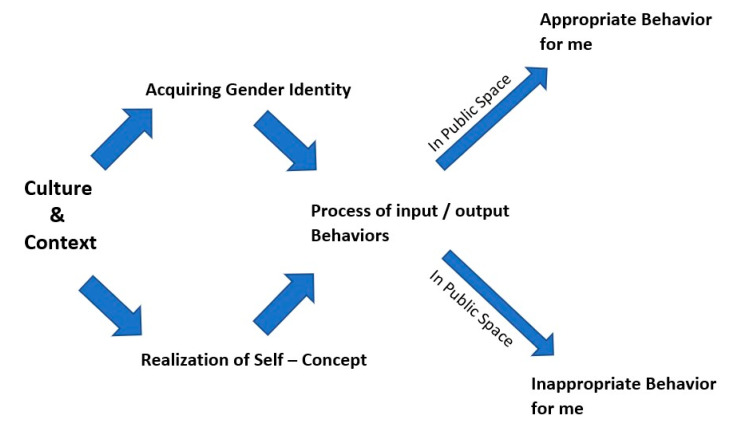
Social cognitive theory in action: a diagram of acquiring appropriate behaviors in public space. Adapted from Shaffer and Kipp [91].

**Figure 4 behavsci-12-00388-f004:**
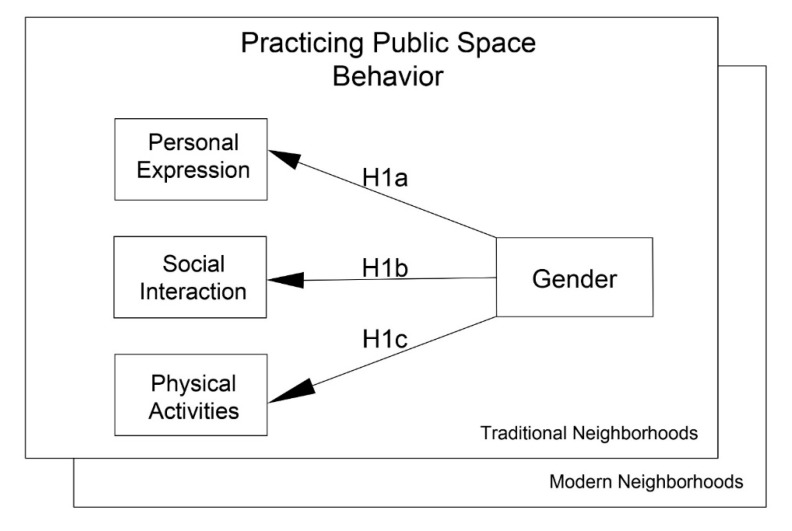
Model 1 for the first three hypotheses.

**Figure 5 behavsci-12-00388-f005:**
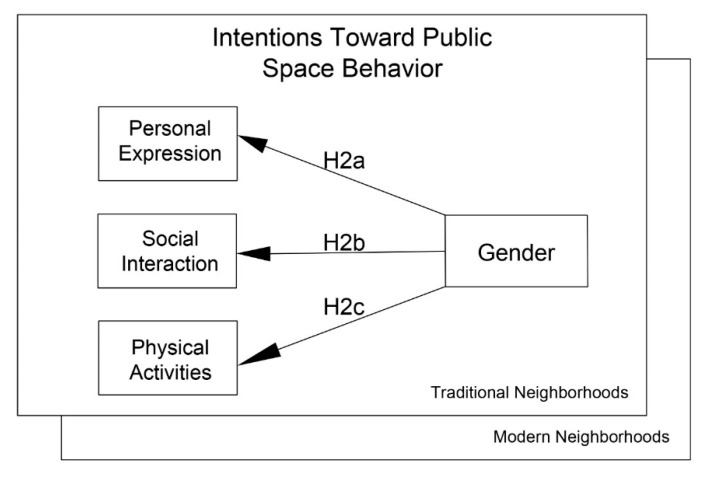
Model 2 for the second three hypotheses.

**Figure 6 behavsci-12-00388-f006:**
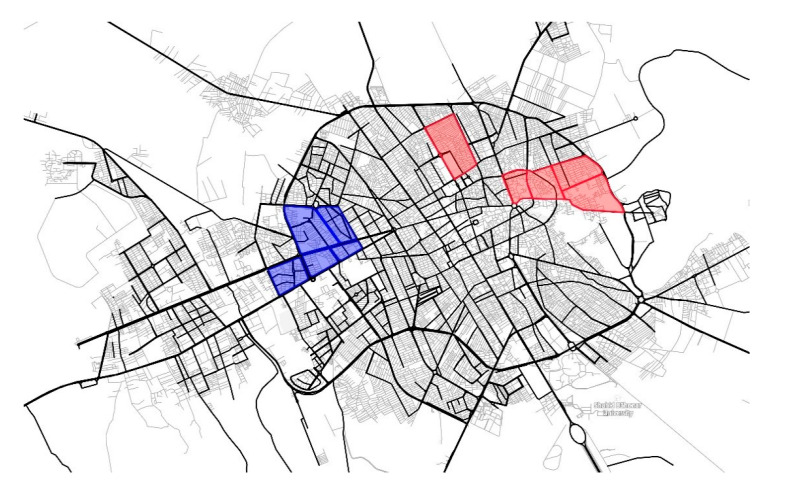
The red areas represent the traditional neighborhoods, and the blue areas represent the modern neighborhoods of the city. (Source: author).

**Figure 7 behavsci-12-00388-f007:**
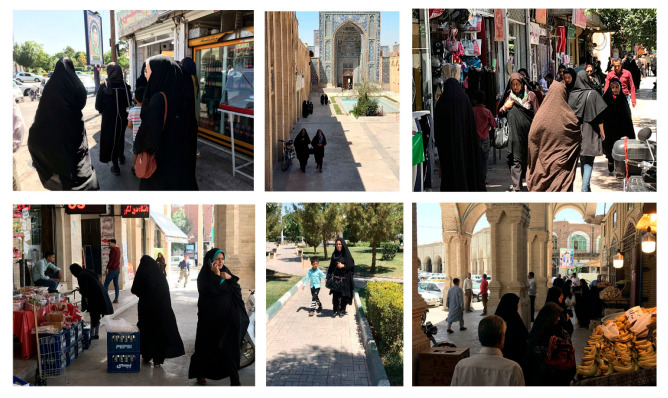
The behaviors in urban public spaces in traditional neighborhoods. (Source author).

**Figure 8 behavsci-12-00388-f008:**
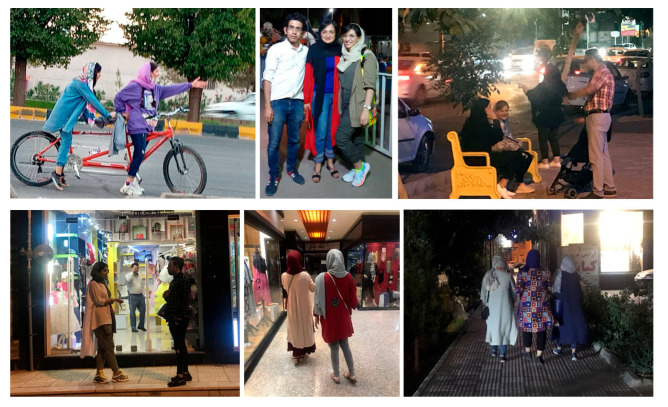
The existing behaviors in urban public space in modern neighborhoods. (Source author).

**Figure 9 behavsci-12-00388-f009:**
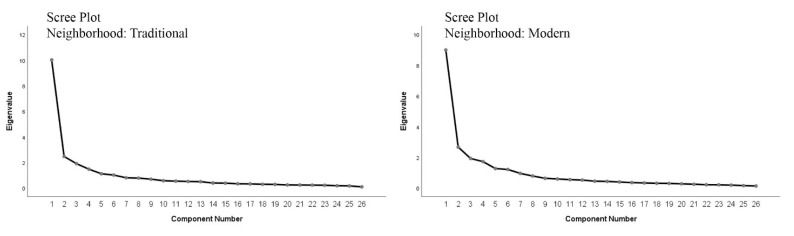
Scree plot of PCA, confirming the six latent variables.

**Table 1 behavsci-12-00388-t001:** Analysis of variance between two districts for both genders.

ANOVA									
		Male				Female			
		Sum of Squares	Mean Square	F	Sig.	Sum of Squares	Mean Square	F	Sig.
Hypothesis 1a: Perceived behavioral restrictions in Personal Expression	Between Groups Modern–traditional	6.269	6.269	6.668	0.011	28.779	28.779	31.940	0.000
	Within Groups	172.974	0.940			211.737	0.901		
	Total	179.2				240.5			
Hypothesis 1b: Perceived behavioral restrictions in Social Interaction	Between Groups Modern–traditional	15.647	15.647	18.507	0.000	34.936	34.936	44.995	0.000
	Within Groups	155.565	0.845			182.467	0.776		
	Total	171.2				217.4			
Hypothesis 1c: Perceived behavioral restrictions in Physical Activities	Between Groups Modern–traditional	0.966	0.966	1.198	0.275	6.713	6.713	8.879	0.003
	Within Groups	147.646	0.807			176.911	0.756		
	Total	148.6				183.6			
Hypothesis 2a: Personal Expressions and level of behavioral appropriateness	Between Groups Modern–traditional	15.741	15.741	17.100	0.000	23.856	23.856	26.810	0.000
	Within Groups	170.298	0.921			209.103	0.890		
	Total	186.0				232.9			
Hypothesis 2b: Social Interactions and the level of behavioral appropriateness	Between Groups Modern–traditional	26.851	26.851	24.342	0.000	33.171	33.171	30.088	0.000
	Within Groups	204.064	1.103			259.078	1.102		
	Total	230.9				292.2			
Hypothesis 2c: Physical Activities and the level of behavioral appropriateness	Between Groups Modern–traditional	2.545	2.545	5.699	0.018	4.683	4.683	8.976	0.003
	Within Groups	82.609	0.447			122.610	0.522		
	Total	85.1				127.2			

**Table 2 behavsci-12-00388-t002:** ANOVA analysis exploring significant gendered behavioral differences in two districts, hypothesis 1a–c.

ANOVA									
		Traditional Neighborhoods				Modern Neighborhoods			
		Sum of Squares	Mean Square	F	Sig.	Sum of Squares	Mean Square	F	Sig.
Hypothesis 1a: Perceived behavioral restrictions in Personal Expression	Between Groups: male–female	8.498	8.498	10.058	0.002	0.333	0.333	0.339	0.561
	Within Groups	163.907	0.845			220.803	0.981		
	Total	172.4				221.1			
Hypothesis 1b: Perceived behavioral restrictions in Social Interaction	Between Groups: male–female	8.207	8.207	11.089	0.001	2.448	2.448	2.833	0.094
	Within Groups	143.577	0.740			194.455	0.864		
	Total	151.7				196.9			
Hypothesis 1c: Perceived behavioral restrictions in Physical Activities	Between Groups: male–female	6.337	6.337	8.054	0.005	1.468	1.468	1.903	0.169
	Within Groups	151.074	0.787			173.483	0.771		
	Total	157.4				174.9			

**Table 3 behavsci-12-00388-t003:** ANOVA analysis exploring significant gendered intention differences in two districts, hypothesis 2a–c.

ANOVA									
		Traditional Neighborhoods				Modern Neighborhoods			
		Sum of Squares	Mean Square	F	Sig.	Sum of Squares	Mean Square	F	Sig.
Hypothesis 2a: Personal Expressions and level of behavioral appropriateness	Between Groups: male–female	0.438	0.438	0.473	0.492	0.058	0.058	0.066	0.798
	Within Groups	179.727	0.926			199.674	0.884		
	Total	180.1				199.7			
Hypothesis 2b: Social Interactions and the level of behavioral appropriateness	Between Groups: male–female	0.932	0.932	0.824	0.365	1.049	1.049	0.973	0.325
	Within Groups	219.380	1.131			243.762	1.079		
	Total	220.3				244.8			
Hypothesis 2c: Physical Activities and the level of behavioral appropriateness	Between Groups: male–female	0.337	0.337	0.544	0.462	0.055	0.055	0.146	0.703
	Within Groups	120.000	0.619			85.219	0.377		
	Total	120.3				85.2			

## Data Availability

The data can be made available upon reasonable request.

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
