# Peer review of "Public Space Behaviors and Intentions: The Role of Gender through the Window of Culture, Case of Kerman"

_behavsci, 2022, doi:10.3390/bs12100388_

Round 1

Reviewer 1 Report

The topic is quite interesting and meaningful. the research design is good enough either.

suggestions:

(1) According to Islamic thought, even men and women are asked to be modest and behave cautiously, but women  carry more of the responsibility (L225-271) , GENDER difference is one of the most important part of the culture. gender and culture in the title looks improper for me.

(2) The second question is about the define of the public space. (LINE 390-414) The questions here are the base of this study, for example , the personal expression assessed questions 2 and three, not to say in Muslin culture, in most urban public space, it is unacceptable to do so for a nomal person.and in line 567 the author mentioned mountains in northern place, but mountain is quite different from urban public space. So I think it is necessarily to define the public space, try to make it more clearly.   and furthemore, I think it better if the authors can explain the process of the sampling, for example, how to find the interviewee, in which places, square, shopping places, park, or libary etc.

(3) L538-L543 IF this finding are substantial finding, it should be emphasize. for example in abstract. 

(4) Line 70-81 are a bit complacent. and new published related papers should be cited.

(5)the discussion and conclusion are wonderful, and I think some new questions are proposed here.

Author Response

We would like to express our gratitude for your kind and encouraging remarks on the paper. We hope that after addressing your comments, the revised version shows significant improvement. We have tried to address your comments in a systematic way in the table below. The manuscript has been submitted with the “track changes” option turned on. Our responses are in the right column, with line numbers as appropriate.

1.      As recommended we have updated the title. The revised phrasing shows that gender is addressed under the umbrella of culture. 

2.      We have tried to provide a clearer definition of public space, and what is considered normal behavior within the specification of the case study in the manuscript (Lines 459-470). The “mountain” mentioned in the manuscript is an extant example in the regional literature (Zakaiee, 2016), it shows how women try to avoid leisurely activities in urban public spaces if possible, the revised version has tried to remove the mentioned ambiguity (Lines 670-674). 

3.      The abstract has been completely reworked addressing your comments and the comments of reviewer 3. 

4.      As recommended by you and other reviewers, we have enlarged our citation and bibliography. The part that you have mentioned has been reworked accordingly (Lines 98-114) The revised version has tried to minimize complacencies and ambiguity. Accordingly, 34 citations have been added throughout the manuscript. 

Thank you. Considering the ongoing situation in Iran, many of these questions need to be further explored.

Reviewer 2 Report

Thank you for a very interesting and important article that addresses an area of urban geography very important to this reviewer (public space, gender and culture). Your contribution has several qualities and I especially appreciate the way you bring together feminist issues and urban geography, claiming a space for their joint positioning so that they can be implemented in policy solutions and decisions. In particular, I think it is a brave article considering the geographic area it focuses on. However, I think your article can be strengthened in several respects and below I offer some suggestions for possible areas of improvement:

1. Consider introducing gender theories in a more comprehensive way at the beginning of the document for readers who are not familiar with this didactic area. Sometimes gender roles are confused with gender stereotypes, and these are different issues that stem from sex-gender theory. This should be better explained. 

2. Consider a more nuanced/reflective approach in which your paper discusses differential socialization and how it influences all educational agents. Differential socialization is a consequence of the naturalization that societies in general have about stereotypes and roles and is a consequence of androcentric and patriarchal culture. I recommend reading on this subject authors such as Marina Subirats, Amparo Tomé, Antonia García Luque, and many others who have coeducation as their line of research.

3. When the document states that the division of labor by sexes occurred in World War II, it is a mistake. This division is previous. On this aspect I recommend reading the book: "The Creation of Patriarchy" by Gerda Lerner.

4. Theoretical framework: I recommend reworking this section. It poses a slightly simplistic (instrumental) view on feminist geography. Within feminist geography there are indeed studies that demonstrate how gender influences the perception and appreciation of public space. For example: 

https://estudiosurbanos.uc.cl/exalumnos/mujeres-ocio-y-apropiacion-del-espacio-publico-una-aproximacion-al-fenomeno-del-ocio-desde-la-geografia-feminista-en-la-ciudad-de-valparaiso/

http://dspace.ucuenca.edu.ec/handle/123456789/32937

5. The Conclusions section needs a better recapitulation of its findings and its contribution to knowledge. In its present form, it is difficult to see how your data support your assertions.  It would be nice if the author could propose lines of improvement for urban public policy.

That said, I enjoyed reading her manuscript and greatly appreciate her ambition to push urban geography research in a much-needed feminist and cultural direction.

Author Response

We would like to express our gratitude for your kind and encouraging remarks on the paper. We hope that after addressing your comments, the revised version shows significant improvement. In general, we were not aware of the rich Spanish literature regarding this topic (due to the language barrier), the current version tries to mitigate that shortcoming by citing the literature recommended by you. We have tried to address your comments in a systematic way in the table below. The manuscript has been submitted with the “track changes” option turned on. Our responses are in the right column, with line numbers as appropriate.

  1. Overall we have tried to better explain this by adding a significant amount of literature review. We have also used the citations that you suggested for this (see Introduction).

2. The influence of educational agents on the development and reinforcement of stereotypes and the socialization process have been addressed in the current version. All recommended scholars have been cited accordingly.

Education (Lines 160-168).

Urban Studies (Lines 46-52).

Feminist Geography (Lines 63-89).

The gap in the literature (Lines 105-115). 

3. Thank you for pointing this out. It has been corrected and cited based on your recommendation (Lines 82-89).

4. We have reworked this part by addressing the recommended literature.

5. Parts of the conclusion have been revised. We have tried to avoid over-generalization due to the specific circumstances of the case study. This has also been added in the end.

Thank you, we appreciate your support. Considering the ongoing situation in Iran, and the struggle for women’s rights, many of these questions need to be further explored in future studies.

Reviewer 3 Report

Dear authors,

I enjoyed reading your manuscript. Investigation of space behavior in a geographical area and the gender-specific issues are highly favorable in gender studies. The manuscript is well-written and structured. Besides, it provides a valid assessment. However, there are some points to take into account in order to develop the paper.

1.      The abstract contains three monotonous sentences of "the study tests, this paper examines, and this study further investigates", even though it could be structured using CARS model suggested by John Swales. Please, reformulate it.

2.      The abstract could be started with the particular territory of your research domain, and the gap that you are filling with this study.

3.      The sentence "implication for research and practice are discussed" is too broad and general, hence does not attract the audience. You may write salient information in your abstract.

4.      Some of the sentences are unnecessarily long and vague, you may break them into robustly meaningful sentences. Take this part of your introduction from line 27 to 33 for example. " Urban studies investigate any differences between men and women’s spatial behavior in urban public spaces and seek explanations. Researchers mostly measure individual’s spatial behavior and types of activities but also look at the concept of space from a social point of view and explore how urban spaces are socially constructed and, consequently, gendered. Urban studies also include the relation between women’s subordination and the levels of their presence and activeness in urban public spaces." Besides, this text includes claims that need to be supported by relevant references.

5.      The paper needs to be revised linguistically by a native or a native-like scholar.

6.      In line 93, your sample is mentioned to be collected for ONE district, while in your abstract you mentioned TWO districts. Please, disambiguate such sentences by including the names of the districts.

7.      Your text from lines 235 to 271 discusses issues related to Islamic cultural effects in gender bias in a narrative sense. You may reformulate non-academic parts, such as sections related to Motahari, Mosavi, and Bagheri to keep the face of your paper more scientific than narrative. Besides, you may also indicate that the definition of Islamic thoughts you have provided in these sections are pro-revolutionary behavioral issues in Iran, which differ from other Islamic areas, synchronically and diachronically.

8.      Please, add data regarding the sampling method you employed.

9.      In line 82, You have mentioned that your research examined how culture and gender shape individual’s public space behavior and the importance they give to different divisions of behaviors in urban public spaces within the framework of social construction theory and social cognition theory. This is a highly interesting investigation, but unfortunately it is not reflected in your discussion and conclusion. Why? In case, you missed the information accidentally, please add it to the conclusion and discussion.

Good luck

Author Response

We would like to express our gratitude for your kind and encouraging remarks on the paper. We hope that after addressing your comments, the revised version shows significant improvement. We have tried to address your comments in a systematic way in the table below. We have reworked the abstract and increased the literature review. The paper has been further proofread, and many ambiguous sentences have been rewritten. The manuscript has been submitted with the “track changes” option turned on. Our responses are in the right column, with line numbers as appropriate.

1. Thank you, the abstract has been fully revised.

2. The abstract has been updated ad re-written to address your comments.

3. The abstract has been updated to address your comments.

4. We have tried to make sentences shorter in the current version and avoid ambiguity. Furthermore, we have significantly increased the citation and bibliography after addressing your comments. In the current revised version, we have tried not to have unsubstantiated claims.

These revisions are scattered throughout the manuscript, for instance (see lines: 45-55; 81-89; 105-115; 160-167; 483-495).

5. The paper has been proofread by a native speaker. The revised version has gone through the proofreading process once more.

6. There are two neighborhoods (one traditional and one modern). These ambiguities have been corrected in the revised version after addressing your comments (Lines 448-452)

7. These parts have been revised and reworked according to your suggestions (Lines 297-320)

8. The sampling method has been added. Necessary citation to the statistic literature has been added accordingly (Lines 453-470)

9. The conclusion has been updated accordingly (Lines 698-715).

Limitations of the study have been also added (Lines 729-732)

Round 2

Reviewer 2 Report

The manuscript has undergone a number of changes that make it a marked improvement over the previous version. 

Congratulations